# Batch and Continuous Column Adsorption of p-Nitrophenol onto Activated Carbons with Different Particle Sizes

Beatriz Ledesma [1],*, Eduardo Sabio [1], Carmen María González-García [1], Silvia Román [1],
Maria Emilia Fernandez [2], Pablo Bonelli [3] and Ana L. Cukierman [3,4]

[1] Departamento de Física Aplicada, Universidad de Extremadura, Avda. Elvas s/n, 06006 Badajoz, Spain; esabio@unex.es (E.S.); cggarcia@unex.es (C.M.G.-G.); sroman@unex.es (S.R.)
[2] Instituto de Desarrollo Tecnológico para la Industria Química (INTEC, UNL-CONICET), Ruta Nacional 168 Km 0, Santa Fe S3000, Argentina; mefernandez@intec.unl.edu.ar
[3] PINMATE, Departamento de Industrias, Instituto de Tecnología de Alimentos y Procesos Químicos ITAPROQ, CONICET, Facultad de Ciencias Exactas y Naturales, Universidad de Buenos Aires, Intendente Güiraldes 2620, Ciudad Universitaria, Buenos Aires C1428BGA, Argentina; pablo@di.fcen.uba.ar (P.B.); analea@di.fcen.uba.ar (A.L.C.)
[4] Cátedra de Farmacotecnia II, Departamento de Tecnología Farmacéutica, Facultad de Farmacia y Bioquímica, Universidad de Buenos Aires, Junín 956, Buenos Aires C1113AAD, Argentina
* Correspondence: beatrizlc@unex.es

**Abstract:** The study focused on investigating the solvent adsorption of p-Nitrophenol (PNP) onto activated carbons for wastewater treatment. It explored the influence of adsorbate concentration and adsorbent size on equilibrium isotherms and removal rates to develop efficient adsorption processes. The study examined adsorption isotherms under equilibrium conditions utilizing both the Langmuir and Double-Langmuir models and the Dubinin–Radushkevich equation. Remarkably, all the models demonstrated equally excellent fitting to the experimental data. Kinetics of PNP adsorption were investigated using pseudo-first-order, pseudo-second-order, and intraparticle diffusion kinetic models. This provided insights into the dominant adsorption mechanism and mass transfer phenomena, aiding the design of efficient wastewater treatment processes. Strong correlations (correlation coefficients > 0.9) were found between the models and experimental data for three types of activated carbons under batch conditions. This validation enhances the reliability and applicability of the models, supporting their practical use. The study also observed a slight increase in maximum adsorption capacity ($q_{max}$) with decreasing particle size, although there is not a significant difference: 340, 350, and 365 mg·g$^{-1}$, for CB-L, CB-M, and CB-S, respectively. This insight helps in selecting appropriate activated carbon for effective PNP removal, considering both adsorption capacity and particle size. Furthermore, the analysis of PNP adsorption under dynamic conditions in fixed-bed columns highlighted the significance of inlet velocity and carbon mass in determining breakthrough time, with particle size playing a secondary role. This information aids in optimizing the design and operation of fixed-bed adsorption systems for efficient PNP removal. In summary, this study's significant contributions lie in enhancing our understanding of PNP adsorption in wastewater treatment. By investigating equilibrium isotherms, kinetics, and mass transfer phenomena, it provides validated models, insights into adsorption capacity and particle size, and practical guidance for dynamic adsorption systems. These findings contribute to the development of efficient and sustainable wastewater treatment methods.

**Keywords:** activated carbons; adsorption; kinetic modelling; p-Nitrophenol

## 1. Introduction

Nowadays, there is increasing concern about the presence of contaminants in drinking water [1,2]. A broadly spread family of contaminants are the phenolic compounds, which are widely used as intermediates in the synthesis of products, such as plastic, dyes,

pesticides, and insecticides [3,4]. Degradation of these substances means the appearance of phenol and its derivatives in the environment [5]. Their presence in drinking water and municipal and industrial waste represents a danger to the environment and, most alarmingly, human health [6,7].

Different techniques can be used to remove phenolic compounds from aqueous solutions, such as chemical oxidation, membrane separation, adsorption, and ion exchange [8]. Adsorption by activated carbons (ACs) has been recognized as an effective process in most industrial water and wastewater treatment. Adsorption of phenolic compounds is still the most investigated of all liquid-phase applications of carbon adsorbents [9,10].

Batch adsorption processes may not be a convenient method for an industrial scale to deal with high-flow rate treatments, although it provides very useful information for the design of industrial-scale processes. Adsorption in a fixed bed column can be used continuously under high effluent flow rates, and it has been used in many pollution control processes [11–14]. However, to properly design and operate fixed-bed filters, the adsorption process must be understood, and both equilibrium and dynamic adsorption studies are required.

The adsorption equilibrium data can be quantitatively interpreted using a wide range of models; for example, the Langmuir model has proved to be a very useful tool in the analysis of adsorption in the liquid phase on activated carbons. However, adsorption in a fixed bed column is a dynamic, non-steady process, which is more difficult to interpret, and the process depends on the possible interaction of several variables [15]. The study of adsorption kinetics is frequently analyzed using dimensionless models (0D) to obtain mathematically sound results [16]. However, these models are often empirical in nature, and their applicability is limited due to their inability to consider the system's geometry and the transport phenomena (mass, momentum, and heat) that occur during the treatment. To improve the understanding of adsorption kinetics and capture the complexities of the system, it is essential to consider specific variables and utilize more comprehensive kinetic models. For example, incorporating variables such as the initial concentration of the adsorbate, contact time, temperature, pH, and the specific surface area of the adsorbent can provide valuable insights [17]. Additionally, advanced kinetic models such as pseudo-first-order, pseudo-second-order, and intraparticle diffusion models can offer a more accurate representation of the adsorption process by considering factors like diffusion limitations, heterogeneous surface reactions, and varying adsorption capacities over time. By employing these specific variables and more comprehensive kinetic models, a more thorough understanding of the adsorption kinetics and its relationship with the system's geometry and transport phenomena can be achieved. For this reason, the 0D model should be used as a starting point for more in-depth analysis using models that consider spatial dimensions (nD models). Although the nD models are more complex than the 0D model, they allow for a deeper understanding of the physical processes taking place during the treatment of contaminated effluents [18,19].

In this paper, the objective was to analyze the adsorption of p-nitrophenol on both batch and continuous column conditions, paying particular attention to the explanation of the mass transfer phenomena by using 2D models.

## 2. Materials and Methods

### 2.1. Materials

A granular activated carbon (CB, from Chemviron Carbon, Beverungen, Germany) was used in this study. The carbon was sieved to different sizes of average diameters of 1.4 mm, 0.29 mm and 0.10 mm, and three fractions were denoted as CB-L, CB-M, and CB-S, respectively. The fractions were used as adsorbents to carry out the adsorption studies. The carbon was dried at 105 °C for 12 h and then stored in a desiccator at room temperature for further use. p-Nitrophenol (PNP) (Sigma, St. Louis, MO, USA, purity > 99%) was used as the adsorbate. Distilled water was used for preparing the solutions.

## 2.2. Characterization of the Adsorbents

The porosity of adsorbents was studied by $N_2$ adsorption at 77 K (AUTOSORB-1, Quantachrome); before analysis, the sample was outgassed at 120 °C for 12 h. Adsorption data were used to calculate typical textural parameters using suitable models. In particular, the following parameters were calculated: (a) BET surface ($S_{BET}$) by BET model [20], (b) micropores volume ($V_{mi}$) through the Dubinin–Radushkevich equation [21], and (c) mesopores volume ($V_{me}$), as the difference between the pore volume at a relative pressure of 0.95 and 0.10. Moreover, the density of the particles ($\rho_p$) was determined by Hg porosimetry (AUTOPORE 4900, Micromeritics, Norcross, GA, USA), and the true density of the sample ($\rho_t$) was estimated using a stereopycnometer (Stereopycnometer SPY-D160-E, Quantachrome, Moscow, Russia) with an accuracy of 0.001 PSI, using helium. From these values, the particle porosity ($\varepsilon$) was calculated.

The Boehm titration method was used to determine the amounts of acidic and basic surface functional groups. This method assumes that NaOH neutralizes carboxyl, lactone, and phenol groups (acidic groups); $Na_2CO_3$ neutralizes carboxyl and lactone; $NaHCO_3$ neutralizes only carboxyl groups, respectively. In addition, HCl neutralizes basic groups [22,23]. Moreover, the point of zero charges (PZC) was estimated to corroborate the results obtained by the Boehm method by mass titrations at different pHs following the procedure described elsewhere [24].

## 2.3. PNP Adsorption Studies

### 2.3.1. PNP Adsorption in Equilibrium Conditions

PNP equilibrium adsorption isotherms were determined based on batch analysis. For this purpose, a fixed amount (0.1 g) of each adsorbent was added to 15 mL of organic solution (adsorbate with distilled water at neutral non-fixed pH because previous works verified the best conditions for the adsorption of this compound [25]) with initial concentrations ($c_0$), ranging 10–100 mg·$L^{-1}$. The flasks were then placed in a thermostatic bath at 293 K and allowed to equilibrate for 48 h since previous experimentation on the adsorption kinetics showed this time was enough to guarantee equilibrium [25].

After the adsorbents were filtered, the concentration in the supernatant solutions ($c_e$) was analyzed by UV-Vis spectrophotometry (spectrophotometer UNICAM Helios-λ) at a wavelength of 225 nm. This wavelength was selected after previous spectral scanning tests, which showed the stability of this signal independently of the possible pH variations. Then, the absorbances of a series of standard solutions of different concentrations were measured by triplication, and the suitability of Beer's law was confirmed. The regression coefficients were very close to one under the range of concentrations used in this study. From the $c_e$ values, the equilibrium solid phase concentration ($q_e$) was obtained.

### 2.3.2. Dynamic PNP Adsorption in Batch Conditions

The kinetic studies were performed, adding 100 mg of each AC to 25 mL of 500 mg·$L^{-1}$ PNP solutions. The flasks were then placed in a thermostatic bath at 293 K with agitation, and the samples were taken at appropriate time intervals. The slurries were filtered through 0.45 μm membranes and PNP concentrations in solution (c), and the corresponding solid-phase concentration (q) values were obtained as described in Section 2.3.1. The tests were carried out at least in duplicate.

### 2.3.3. Dynamic PNP Adsorption in Fixed-Bed Columns

For continuous adsorption experiments, an acrylic column (1.6 cm internal diameter and 34 cm height) was packed with a different mass of adsorbent (3 and 6 g) of an average particle diameter of 0.29 mm and 0.10 mm. Prior to the measurements, the bed was washed with distilled water to remove all interstitial air. To complete the volume of the column, glass beads were also packed, and metallic sieves in the extremes were used to avoid losses of material.

The column was then fed with a 2000 mg·L$^{-1}$ of PNP solution. Two flow rates (5.5 and 11 cm$^3$·min$^{-1}$) were investigated. A peristaltic pump (Masterflex1–100 rpm, Cole–Parmer, Vernon Hills, IL, USA) with a flow controller was used. Samples were collected at the exit of the column at predetermined time intervals using a solenoid valve and an automatic sample collector; the PNP concentrations were analyzed spectrophotometrically in triplicate. As earlier described, all the experiments were carried out at 293 K using a thermostatic system (Lauda Ecoline E 300, Lauda Dr. R. Wobser GMBH & CO.KG, Lauda-Königshofen, Germany). The temperature was further monitored by an internal thermocouple connected to a digital controller (Iea, Micro 80, Argentine Electronic Engineering S.R.L., Rosario, Argentina). Figure 1 shows a basic scheme of the experimental setup.

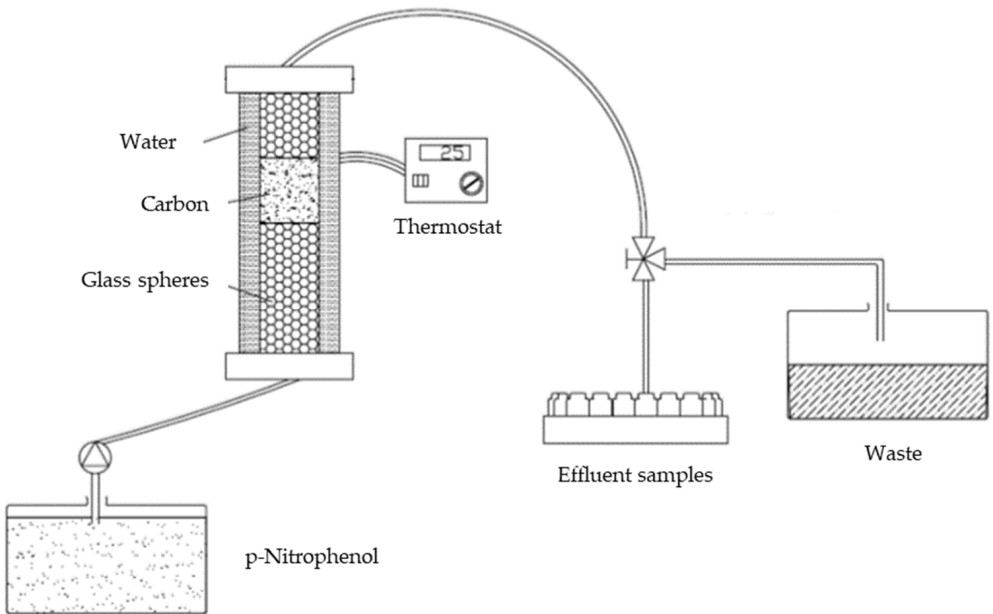

**Figure 1.** Scheme of the experimental installation for column adsorption.

## 3. Results

### 3.1. Characterization of the Adsorbents

From nitrogen adsorption, typical AC adsorption parameters were determined: BET surface (S$_{BET}$), volume of micropores (V$_{mi}$), and volume of mesopores (V$_{me}$), and are collected in Table 1.

As can be seen, the three samples have similar porosity patterns, and there are no significant differences among the three carbons in both V$_{mi}$ and V$_{me}$. As for the densities, the three samples show a similar pattern again, with particle density ($\rho_p$) and true density ($\rho_t$) values being very similar. However, the samples show important differences in their geometrical characteristics. The average particle diameters ($\Phi_p$) were 1.40 mm, 0.29 mm, and 0.10 mm for CB-L, CB-M, and CB-S, respectively. Assuming a spherical shape for the particles, the corresponding particle surface (S$_p$), particle volume (V$_p$), and the ratio between both were calculated (S$_p$/V$_p$). It is interesting to note the significant difference in the S$_p$/V$_p$ ratio between the samples because this ratio plays an important role in the kinetics of the adsorption process. Sabio et al. [19] analyzed seven different carbons founding that the rate of the adsorption process depends to a large extent on the S$_p$/V$_p$ ratio and existing a negative linear relationship between both parameters.

Acid and basic sites and PCZ were quantified, and the results are shown in Table 1, where the number of basic sites on the three ACs is significantly higher compared to the number of acid sites, and the PCZ is 10.2, indicating the overall basic character of this activated carbon.

**Table 1.** Textural and geometrical characteristics of the carbons.

| Parameter | CB-L | CB-M | CB-S |
|---|---|---|---|
| $S_{BET}$ ($m^2 \cdot g^{-1}$) | 930 | 861 | 893 |
| $V_{mi}$ ($cm^3 \cdot g^{-1}$) | 0.490 | 0.438 | 0.454 |
| $V_{me}$ ($cm^3 \cdot g^{-1}$) | 0.063 | 0.058 | 0.061 |
| $\rho_p$ ($g \cdot cm^{-3}$) | 1.0241 | 1.0209 | 1.1283 |
| $\rho_t$ ($g \cdot cm^{-3}$) | 2.138 | 2.138 | 2.127 |
| $\varepsilon$ | 0.521 | 0.522 | 0.470 |
| $\Phi_p$ (mm) | 1.40 | 0.29 | 0.10 |
| $S_p$ ($mm^2$) | 6.16 | 0.26 | $3.14 \times 10^{-2}$ |
| $V_p$ ($mm^3$) | 1.44 | $1.28 \times 10^{-2}$ | $5.24 \times 10^{-4}$ |
| $S_p/V_p$ ($mm^{-1}$) | 4.29 | 20.69 | 60.00 |
| $m_p$ (mg) | 1.47 | $1.30 \times 10^{-2}$ | $5.91 \times 10^{-4}$ |
| Carboxyl groups ($mEq \cdot g^{-1}$) | 0.02 | 0.02 | 0.02 |
| Lactone groups ($mEq\ g^{-1}$) | 0.01 | 0.01 | 0.01 |
| Phenol groups ($mEq\ g^{-1}$) | 0.13 | 0.13 | 0.13 |
| Carbonil groups ($mEq\ g^{-1}$) | 0.07 | 0.07 | 0.07 |
| Total surface acidic groups ($mEq\ g^{-1}$) | 0.23 | 0.23 | 0.23 |
| Total surface basic groups ($mEq\ g^{-1}$) | 0.39 | 0.39 | 0.39 |
| Point Zero Charge | 10.2 | 10.2 | 10.2 |

*3.2. PNP Adsorption Studies*

3.2.1. PNP Adsorption in Equilibrium Conditions

As a first step, the study of the PNP adsorption process was carried out under equilibrium conditions. Figure 2 shows the equilibrium adsorption isotherms ($q_e$ vs. $c_e$) obtained for the three samples and describes the relationship between the concentration of adsorbate and the amount of adsorbate adsorbed onto an adsorbent at a given temperature and pressure. The phenomena responsible for these isotherms include adsorption and desorption, surface coverage, equilibrium, adsorption capacity, and the type of isotherm observed. It can be observed that in all cases, there is an initial section almost vertical up to $c_e \sim 15$ mg$\cdot$L$^{-1}$ and $q_e \sim 200$ mg$\cdot$g$^{-1}$; then, as $c_e$ increases, the slope of the isotherms decreases, although a plateau is not reached in the range of concentration used in this study. Thus, the three isotherms can be classified as type L according to the Giles classification [26]. The adsorption isotherms have been adjusted by the Langmuir and Double-Langmuir models, which generally describe L-class isotherms accurately [27]. The equations of these isotherms are:

$$\text{Langmuir isotherm}: q = q_L \frac{b \cdot c}{1 + b \cdot c} \tag{1}$$

$$\text{Doble Langmuir isotherm } q = q_{L,1} \frac{b_1 \cdot c}{1 + b_1 \cdot c} + q_{L,2} \frac{b_2 \cdot c}{1 + b_2 \cdot c} \tag{2}$$

where q is the amount of adsorbate adsorbed on the surface (mg$\cdot$g$^{-1}$); $q_L$, Langmuir maximum adsorption capacity (mg$\cdot$g$^{-1}$); b, Langmuir constant (L$\cdot$mg$^{-1}$); c, the equilibrium concentration of the adsorbate in the liquid (mg$\cdot$L$^{-1}$); $q_{L,1}$, Langmuir maximum adsorption capacity associated with the first adsorption layer in the double Langmuir model (mg$\cdot$g$^{-1}$); $q_{L,2}$, Langmuir maximum adsorption capacity associated with the first adsorption layer in the double Langmuir model (mg$\cdot$g$^{-1}$); $b_1$: the equilibrium constant associated with the first adsorption layer in the double Langmuir model (L$\cdot$mg$^{-1}$); $b_2$: the equilibrium constant associated with the second adsorption layer in the double Langmuir model (L$\cdot$mg$^{-1}$).

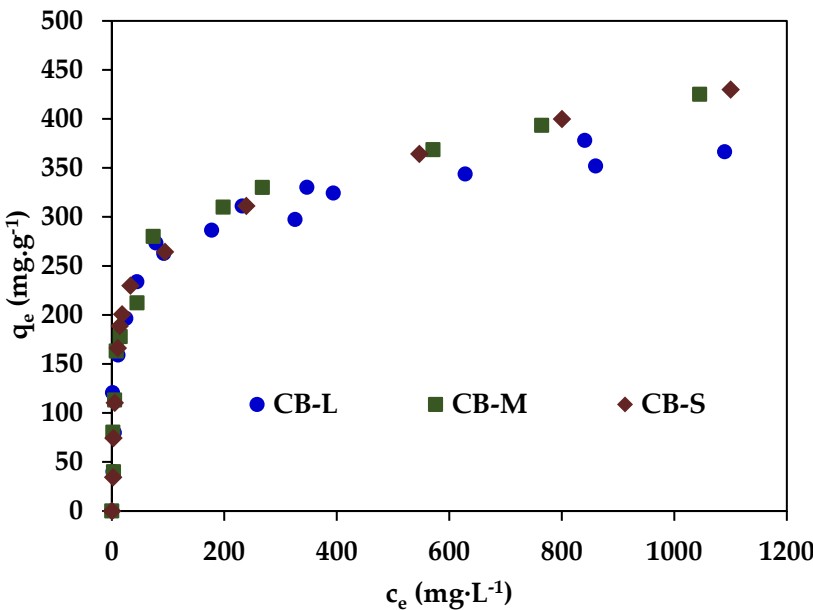

**Figure 2.** PNP equilibrium adsorption isotherms for CB-L, CB-M, and CB-S.

The results obtained from this analysis are listed in Table 2, where it can be observed from the regression coefficient $R^2$ values that both models demonstrate an excellent fit to the data.

**Table 2.** Results from analysis by Langmuir and Double-Langmuir models of PNP equilibrium adsorption isotherms.

| | Langmuir | | | Double-Langmuir | | | | |
|---|---|---|---|---|---|---|---|---|
| Sample | $q_L$, (mg·g$^{-1}$) | $b$, (L·mg$^{-1}$) | $R^2$ | $q_{L1}$, (mg·g$^{-1}$) | $b_1$, (L·mg$^{-1}$) | $q_{L2}$, (mg·g$^{-1}$) | $b_2$, (L·mg$^{-1}$) | $R^2$ |
| CB-L | 343.7 | 0.0582 | 0.9281 | 250.1 | 0.178 | 435.8 | $4.3 \times 10^{-4}$ | 0.9540 |
| CB-M | 374.1 | 0.0580 | 0.9382 | 280.0 | 0.130 | 447.2 | $4.0 \times 10^{-4}$ | 0.9816 |
| CB-S | 377.3 | 0.0628 | 0.9449 | 285.1 | 0.134 | 439.0 | $4.9 \times 10^{-4}$ | 0.9937 |

Based on the results obtained in Table 2 can be inferred that the higher adsorption capacity observed for the smallest particle size (CB-S) can be attributed to smaller particles generally possessing a higher surface area/volume ratio compared to larger particles (CB-L). This increased surface area provides more active sites for adsorption, allowing for greater interaction between the adsorbate molecules and the activated carbon surface. Furthermore, the diffusion limitations are reduced in smaller particles due to their shorter diffusion pathways. This enables more efficient transport of the adsorbate molecules from the bulk solution to the active sites on the activated carbon surface, leading to enhanced adsorption capacity.

The isotherm data obtained were also analyzed using the Dubinin–Radushkevich (D-R) isotherm model (Equation (3)) [28]. This model assumes that the adsorption is related to surface porosity and pore volume.

$$q = q_{DR} \cdot exp \left[ - \left( \frac{RT ln\left(1 + c^0/c\right)}{E_{DR}} \right)^2 \right] \tag{3}$$

where $q_{DR}$ represents the maximum adsorption capacity according to the D-R model (mg·g$^{-1}$), $T$ is the equilibrium temperature (K), $R$ is the gas constant (kJ·mol$^{-1}$K$^{-1}$), $c^0$ is

the standard state for the solution (1 M), $c$ is in molar units, and $E_{DR}$ is the D-R energy adsorption (kJ·mol$^{-1}$).

The D-R isotherm model examines adsorption from an energetic point of view and specifies that the adsorption process is physical or chemical [29]. If the $E_{DR}$ value is in the range 8–16 kJ mol$^{-1}$, the adsorption process is chemical, if the $E_{DR}$ value is <8 kJ mol$^{-1}$, the adsorption process is physical. Table 3 shows the parameters of the D-R Equation (3) for the three samples. By analyzing the $E_{DR}$ value, which quantifies the energy of the adsorption process, we can determine that the adsorption in our systems is physical, which represents a lower energy range.

**Table 3.** Parameters of the D-R Equation (6) for the three samples.

| | Langmuir | | |
|---|---|---|---|
| **Sample** | $q_{RD}$, (mg·g$^{-1}$) | $E_{DR}$, (kJ·mol$^{-1}$) | $R^2$ |
| CB-L | 888.1 | 3.8 | 0.9504 |
| CB-M | 1089.7 | 3.5 | 0.9449 |
| CB-S | 1232.9 | 3.4 | 0.9500 |

### 3.2.2. Dynamic PNP Adsorption in Batch Conditions

Figures 3 and 4 show the kinetic curves of the PNP concentration in bulk liquid (c) and the PNP adsorbed in the solid (q), respectively, for the three samples studied. Even though the three ACs show very similar properties, they have different adsorption kinetics of PNP, mainly CB-L. This fact indicates that the size of the particles plays an important role in the kinetic behavior. A sharp drop in the concentration of solute in the bulk solution can be observed with a concomitant sharp increase of adsorption in the solid particle. The sample with a greater size (CB-L) adsorbs at a lower rate than the other samples. This fact is most likely related to its smaller surface/volume ratio, $S_p/V_p$ (Table 1). In samples CB-M and CB-S, the mass transfer rate of PNP from the solution to the particle decreases drastically around 20 min, and the equilibrium is reached at ~40 min (horizontal plateau). In sample CB-L, the reduction of the mass transfer rate is more gradual, and the equilibrium is not reached until ~100 min.

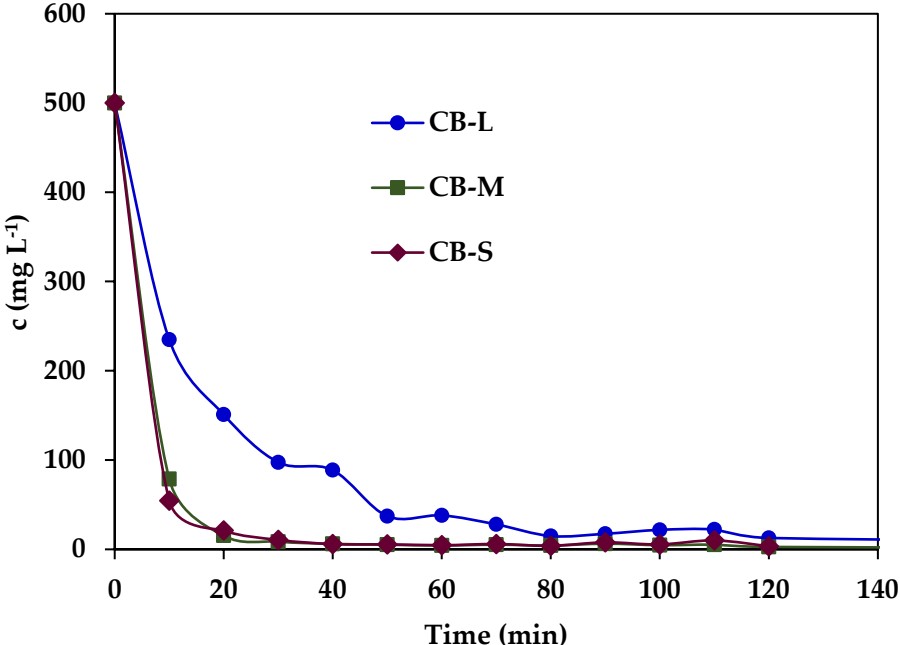

**Figure 3.** Variation of PNP concentration (c) with time in the bulk solutions.

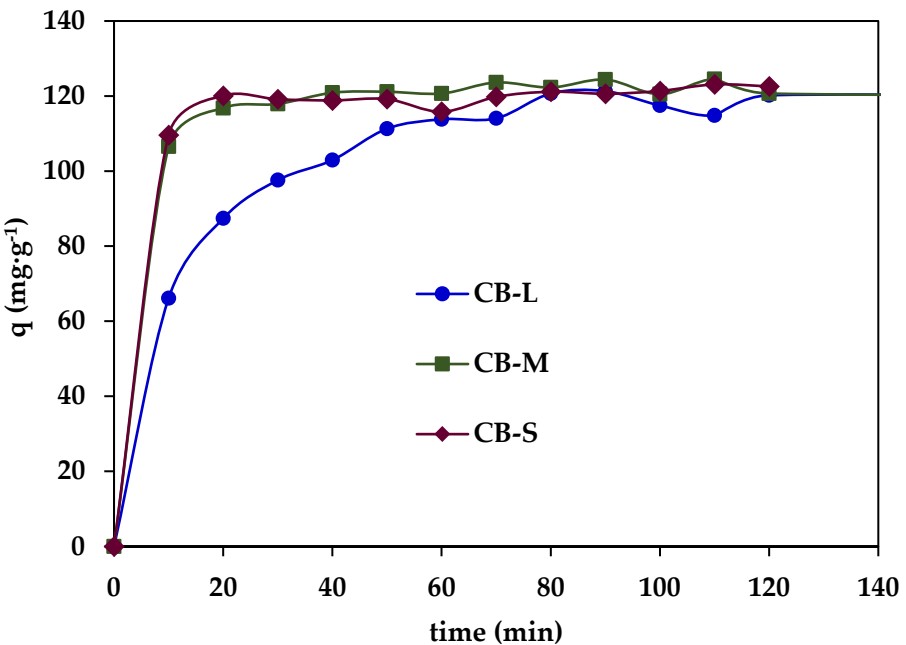

**Figure 4.** Variation of PNP adsorbed (q) with time.

The adsorption process is generally described through four consecutive steps: (1) transport of solute into the bulk solution; (2) film diffusion from the bulk solution through the adsorbent liquid boundary layer; (3) intraparticle diffusion in the adsorbent pores; and (4) adsorption reactions on the active sites of the surface [30,31]. In our case, step 1 can be neglected because the liquid was stirred, and the concentration of solute can be assumed homogeneous in the bulk liquid. The kinetics of PNP adsorption was modeled using the pseudo-first-order (PFO), pseudo-second-order (PSO), and intraparticle diffusion kinetic models to determine the dominating adsorption mechanism [32]. PFO assumes that physisorption limits the rate of adsorption of the solute particles onto the adsorbent, while the PSO model considers chemisorption as the rate-limiting mechanism of the process. Physical adsorption, or physisorption, is caused by Van der Waals or electrostatic forces, while chemisorption involves the formation of chemical bonds between the adsorbent and the solute [33]. The PFO and PSO equations are:

$$\text{PFO model}: q = q_e \left( 1 - e^{-K_1 t} \right) \tag{4}$$

$$\text{PSO model}: \ q = \frac{K_2 q_e^2 t}{1 + K_2 q_e t} \tag{5}$$

where $q$ and $q_e$ (mg·g$^{-1}$) are the sorption capacity at a time $t$ and equilibrium, respectively, while $K_1$ (min$^{-1}$) and $K_2$ (g mg$^{-1}$min$^{-1}$) are the rate constants of the pseudo-first and pseudo-second-order models, respectively [34]. Table 4 shows the resulting PFO and PSO models parameters. It can be observed that both models fit the experimental data, with correlation coefficients ($R^2$) ranging from 0.9819–0.9970 and 0.9890–0.9968, respectively.

Moreover, the three samples show similar values of the experimental equilibrium adsorption capacities ($q_e$) (122.4, 121.4, and 121.8, for CB-L, CB-M, and CB-S, respectively), and the calculated $q_e$ of both the PFO and PSO models. The goodness of fit of the experimental values by both models may suggest that physical and chemical interactions may simultaneously contribute and control the uptake of PNP onto the three AC surfaces. Finally, it can be observed that the values of both $K_1$ and $K_2$ increase with decreasing particle size. Thus, $K_1$ is 0.064, 0.205, and 0.243 (min$^{-1}$), and $K_2$ is $9.91 \cdot \times 10^{-4}$, $6.27 \cdot \times 10^{-3}$, and $7.97 \cdot \times 10^{-3}$ (g mg$^{-1}$min$^{-1}$) for CB-L, CB-M, and CB-S, respectively. This pattern in the kinetic values agrees with the evolution observed in Figures 3 and 4.

**Table 4.** PFO and PSO model parameters with standard errors for PNP adsorption on CB-L, CB-M, and CB-S.

| Sample | PFO Model | | | | PSO Model | | |
|---|---|---|---|---|---|---|---|
| | $q_e$ (exp) (mg·g$^{-1}$) | $q_e$ (mg·g$^{-1}$) | $K_1$ (min$^{-1}$) | $R^2$ | $q_e$ (mg·g$^{-1}$) | $K_2$ (g mg$^{-1}$ min$^{-1}$) | $R^2$ |
| CB-L | 122.4 | 119.7 ± 1 | 0.064 ± 0.04 | 0.9819 | 125.6 ± 1 | $9.91 \times 10^{-4} \pm 8 \times 10^{-5}$ | 0.9890 |
| CB-M | 121.4 | 121.4 ± 0.5 | 0.205 ± 0.01 | 0.9950 | 123.1 ± 0.7 | $6.27 \cdot \times 10^{-3} \pm 1 \times 10^{-3}$ | 0.9940 |
| CB-S | 121.8 | 120.2 ± 0.6 | 0.243 ± 0.02 | 0.9970 | 122.4 ± 0.9 | $7.97 \cdot \times 10^{-3} \pm 1 \times 10^{-3}$ | 0.9968 |

To determine if diffusion mechanisms affect the adsorption of PNP onto the ACs, the Weber–Morris intraparticle diffusion model [35] was evaluated:

$$\text{Weber–Morris model :} \quad q = k_{ID}t^{0.5} + C_{ID} \qquad (6)$$

where $k_{ID}$ (mg g$^{-1}$ min$^{-0.5}$) is the intraparticle diffusion constant, $t^{0.5}$ is the square root of time (min), and $C_{ID}$ (mg g$^{-1}$) is related to the thickness of the boundary layer. The model assumes that the adsorption process is controlled by the transport of the solute molecules from the bulk solution into the internal pores of the adsorbent, and a plot of q versus $t^{0.5}$ provides information on the diffusion mechanisms influencing the adsorption process. Table 5 lists the results obtained from the application of the Weber–Morris model to the adsorption of PNP onto CB-L, CB-M, and CB-S. In sample CB-L two distinct linear segments were observed (line 1: t = 0–20 min and line 2: t = 20–60 min), which indicates the existence of several diffusion steps. Line 1 corresponds to the fast initial sorption. It may be primarily attributed to the instantaneous or very rapid sorption on the external surface of carbons that provides readily accessible sorption sites for PNP, which is controlled by the liquid film diffusion of the solute. Line 2 corresponds to the intraparticle diffusion into the CB-L pores.

**Table 5.** Weber–Morris model parameters with standard errors for PNP adsorption on CB-L, CB-M, and CB-S.

| Sample | Line 1: t = 0–20 min | | | Line 2: t = 20–60 min | | |
|---|---|---|---|---|---|---|
| | $k_{ID,1}$ (mg g$^{-1}$ min$^{-0.5}$) | $C_{ID,1}$ (mg g$^{-1}$) | $R^2$ | $k_{ID,2}$ (mg g$^{-1}$ min$^{-0.5}$) | $C_{ID,2}$ (mg g$^{-1}$) | $R^2$ |
| CB-L | 21.3 | 1.51 | 0.9786 | 7.6 | 56.1 | 0.9719 |
| CB-M | 7.8 | 81.8 | 1 | -- | -- | -- |
| CB-S | 8.0 | 84.3 | 1 | -- | -- | -- |

The higher value of the rate constant for the first step ($k_{ID,1}$) than the rate constant of the second step ($k_{ID,2}$) suggests that the diffusion resistance across the liquid boundary layer is smaller compared to the resistance existing during the pore diffusion step. CB-M and CB-S uptake showed only a linear segment, with very similar $k_{ID,1}$ and $C_{ID}$ values for both samples. The value of $k_{ID,1}$ for CB-M and CB-S is about half of that found in CB-L. On the other hand, it should be noted that $C_{ID}$ for CB-M and CB-S is much higher than CB-L, suggesting that the boundary layer effect is larger for both CB-M and CB-S as compared to CB-L adsorption.

The application of the above models supplies useful information about the kinetics of PNP adsorption on the studied samples. Nevertheless, although the information provided by these models is useful, these 0D models do not take into account the effect of spatial dimensions on kinetic adsorption. To get a deeper insight into the mass transfer phenomena taking place, a 2D model has been applied. The particle was considered to be spherical for the three carbons. The mass balance in particles was defined by Equation (7):

$$\varepsilon \frac{\partial c}{\partial t} + \rho_p \frac{\partial q}{\partial t} + \nabla \left( -D_{F,p} \nabla c \right) = 0 \tag{7}$$

where $\varepsilon$ is the particle porosity, $\rho_p$ (g cm$^{-3}$) the particle density, and $D_{F,p}$ (mm$^2$·s$^{-1}$) the diffusivity of the solute throughout the fluid inside the pores of the particle. The driving force throughout the boundary layer was set to be proportional to the difference between the solute concentration of the bulk liquid ($c_{bulk}$) and of the liquid into the porosity at the surface of the solid ($c_{solid}$) (Equation (8)):

$$\dot{m}_{BL} = k_{BL} \cdot (c_{bulk} - c_{solid}) \tag{8}$$

where $\dot{m}_{BL}$ (mg s$^{-1}$) is the mass flow throughout the boundary layer, and $k_{BL}$ (mm·s$^{-1}$) is the mass transfer coefficient in the boundary layer. Once the solute adsorbed on the porous matrix of the particle, it is assumed that the solute diffuses through the particle pores following Fick's Law, while an instantaneous equilibrium between the solute in the solution and that attached to the porous walls is achieved [36,37]. According to that, the mass balance of solute over a volume element of the carbon particle is defined by Equation (7), which states that the rate of change of the solute mass inside a volume element, both dissolved and attached, equals the rate of mass diffusion. In order to resolve this equation, the relationship between c and q was established. In this case, it was supposed that equilibrium was defined by the Langmuir isotherm (Equation (1)). Thus, the relationship between c and q was defined from the derivation of the Langmuir isotherm (Equation (9)):

$$\frac{\partial q}{\partial c} = q_{max} \cdot \frac{b \cdot q_{max}}{(1 + b \cdot c)^2} \tag{9}$$

The model was solved by using software of finite elements COMSOL Multiphysics. Table 6 shows the values of the model parameters, as well as the correlation coefficients obtained for the fitting of the kinetic concentration values c in the bulk solution ($R_c^2$) and q in the solid ($R_q^2$). The high values of both coefficients, as well as the lines in Figures S1–S4 in the Supplementary Information, put evidence that the model fits experimental data from the experiments carried out under batch conditions for the three ACs. It should be noted that, although in most papers, only a regression coefficient is calculated ($R_c^2$ or $R_q^2$) [38,39], these two coefficients must be calculated in order to verify the validity of the model because the prediction of the kinetics of both q and c must be consistent with the global mass balance.

**Table 6.** Parameters of the 2D-batch model.

| Parameter | CB-L | CB-M | CB-S |
|:---:|:---:|:---:|:---:|
| $q_{max}$ (mg·g$^{-1}$) | 340 | 350 | 365 |
| $b$ (L·mg$^{-1}$) | 0.0142 | 0.0510 | 0.0600 |
| $D_{F,p}$ (mm$^2$·s$^{-1}$) | $12.0 \times 10^{-2}$ | $8.0 \times 10^{-2}$ | $4.5 \times 10^{-2}$ |
| $k_{BL}$ (mm·s$^{-1}$) | $10.5 \times 10^{-2}$ | $5.4 \times 10^{-2}$ | $2.7 \times 10^{-2}$ |
| $R_c^2$ | 0.9498 | 0.9963 | 0.9987 |
| $R_q^2$ | 0.9916 | 0.9925 | 0.9956 |

The values of the maximum adsorption capacity ($q_{max}$) obtained increased with the decreasing size of the particle, although there is not a significant difference: 340, 350, and 365 mg·g$^{-1}$ for CB-L, CB-M, and CB-S, respectively. These values are close to the experimental values (see Table 2). The parameter b of Langmuir also increases as the particle size decreases: 0.0142, 0.0510, and 0.06 (L·mg$^{-1}$) for CB-L, CB-M, and CB-S, respectively. These values indicate that, according to the 2D-Batch model in CB-L, the interaction between the solute and the carbon surface is weaker than that in CB-M and CB-S. The value of

the mass transfer coefficient in the boundary layer ($k_{BL}$) and the diffusivity of the solute throughout the fluid inside the pores of the particle ($D_{F,p}$) drop with the reducing size:

$$k_{BL,CB-L} = 10.5 \times 10^{-2} \ (mm \ s^{-1}) > k_{BL,CB-M} = 5.4 \times 10^{-2} \ (mm \ s^{-1}) > k_{BL,CB-S} = 2.7 \times 10^{-2} \ (mm \ s^{-1})$$

$$D_{F,p,CB-L} = 12.0 \times 10^{-2} \ (mm^2 \ s^{-1}) > D_{F,p,CB-M} = 8.0 \times 10^{-2} \ (mm^2 \ s^{-1}) > D_{F,p,CB-S} = 4.5 \times 10^{-2} \ (mm^2 \ s^{-1})$$

This fact indicates that in sample CB-L, the solute diffuses faster in both the boundary layer and the liquid inside the pores. The consequences of the pattern observed for the values of $k_{BL}$ and $D_{F,p}$ can be observed in Figure 5a–c, where the solution concentration versus the dimensionless radius $r/r_p$ ($r_p$ = particle radius) at 10 min, 20 min, and 50 min is plotted. It can be observed that, since the beginning, c is much higher in the outer part of the particle in sample CB-L than in samples CB-M and CB-S, indicating that the solute diffuses very fast through the pores of the bigger sample (CB-L). The reason for that, as can be inferred from Figure 6a–c, where the values $c \cdot c_0^{-1}$ at the bulk liquid and $q \cdot q_{max}^{-1}$ at the particle have been collected at 10 min, 20 min, and 50 min for the three samples. Sample CB-L presents a high concentration, near saturation ($q \cdot q_{max}^{-1} \sim 1$) at the outer core of the particle. As a result, the solute that enters the pores is in contact with a pore surface that is almost saturated, flowing and dissolved into the inner part of the particle. In contrast, in sample CB-S, the solute dissolved in the pores is in contact with a pore surface with a low number of molecules adsorbed ($q \cdot qmax^{-1} << 1$); thus, under these conditions, the local equilibrium (described by the Langmuir isotherm) favors the adsorption of the solute onto the free surface of the pore, which is removed from the liquid of the pore. As a result, inside the smaller particles, the gradients in both c and q have been minimal since the beginning of the process (see Figures 5a–c and 6a–c). For example, at 20 min in the CB-S experiment, the bulk solution is close to $c_e$, and the particle presents an even distribution of q, with values near $q_e$. At 50 min, according to the model, sample CB-S is clearly at equilibrium because there is no gradient of q inside the particle, and c is $c_e$ in the whole fluid (bulk and pores). On the contrary, for CB-L, the gradients of c and q inside the particle remain the same, even at 50 min. The behavior of CB-M is like that of CB-S. These facts agree with the results found in the application of the Weber–Morris model, where sample CB-L showed two segment lines (line 1: t = 0–20 min and line 2: t = 20–60 min), while CB-M and CB-S showed only one (line 1: t = 0–20 min).

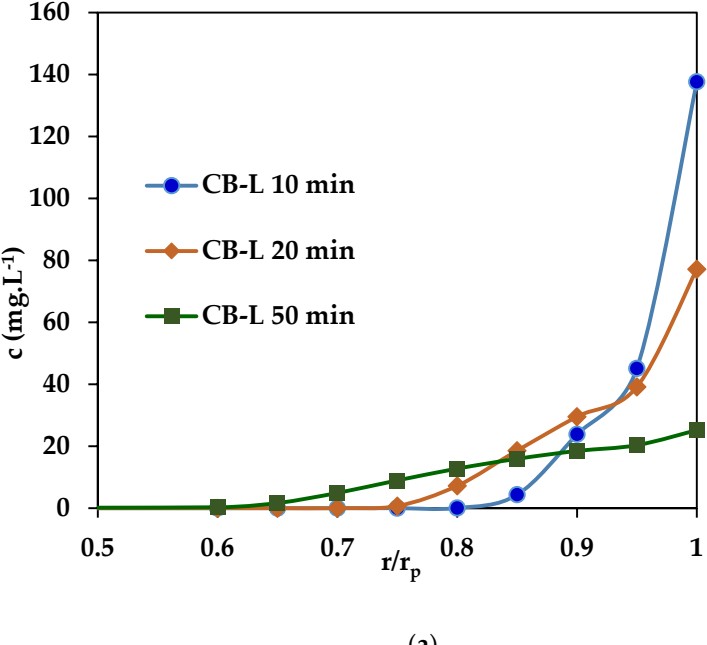

(a)

**Figure 5.** *Cont.*

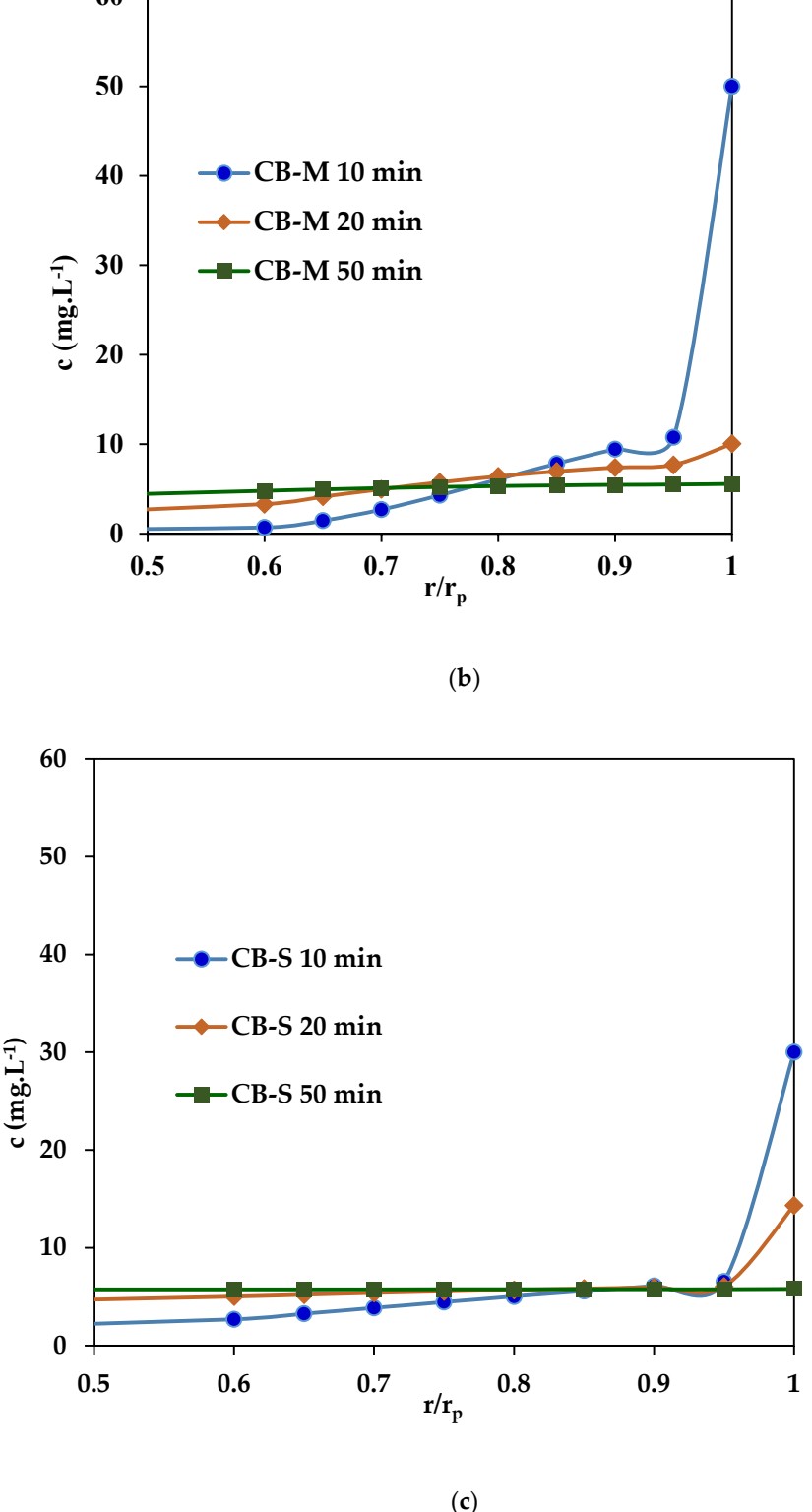

(**b**)

(**c**)

**Figure 5.** Profile of the c values along the dimensionless radius, $r/r_P$, at 10 min, 20 min, and 50 min. (**a**) CB-L; (**b**) CB-M.; (**c**) CB-S.

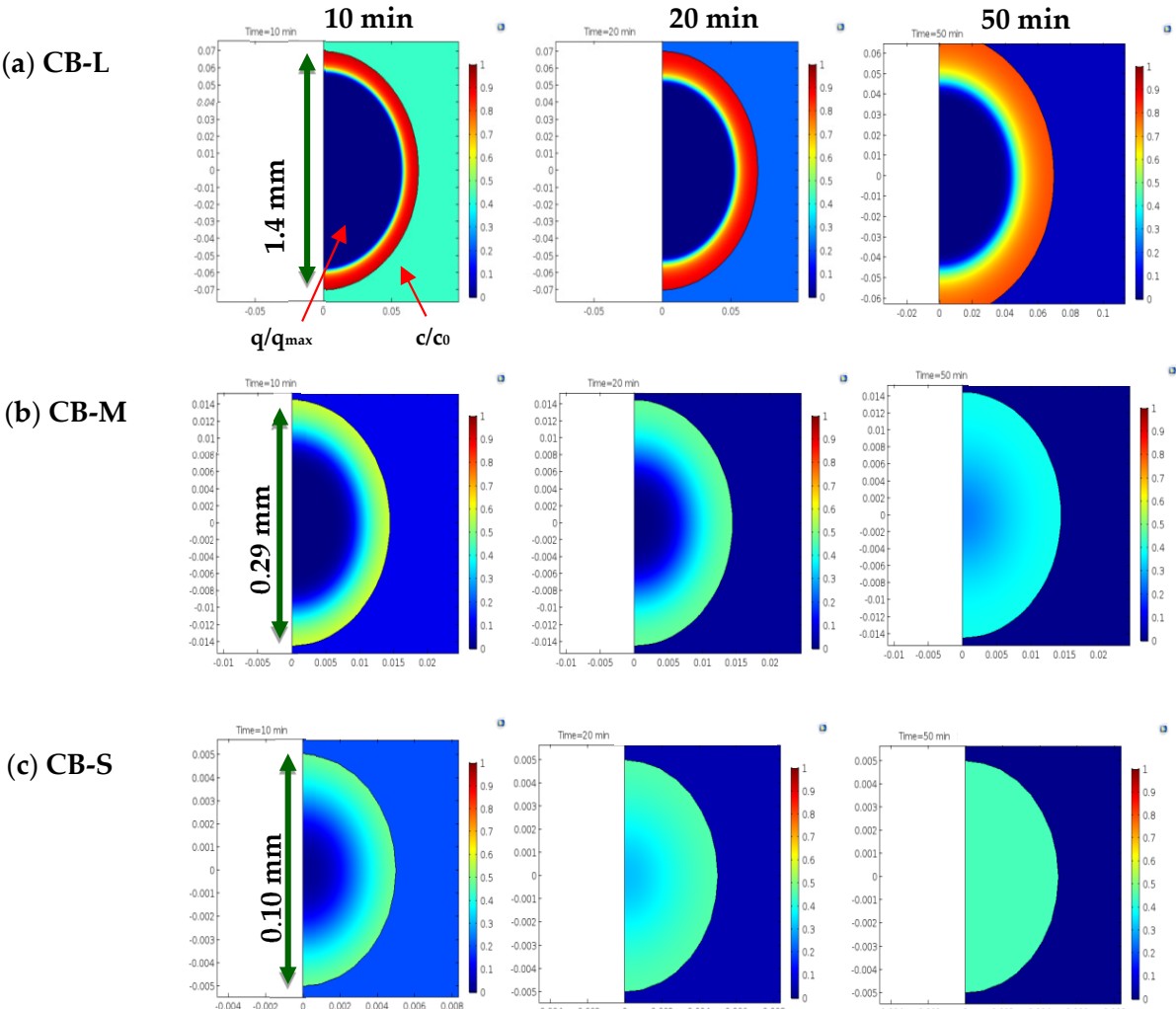

**Figure 6.** 2D plots of the $c \cdot c_0^{-1}$ and $q \cdot q_{max}^{-1}$ values at 10 min, 20 min, and 50 min for the three samples of AC.

The presence of the saturated outer-layer zone, with high q and c values, hinders the mass transfer process of the solute from the bulk solution into the solid particle, slowing down the removal of PNP from the bulk solution. Despite the three samples have similar textural and porosity properties, they present very different adsorption kinetics, which puts in evidence that the particle size plays a main role in the rate of the mass transfer process. The behavior observed in the PNP adsorption kinetics is most likely related to the $S_p/V_p$ of the three samples. It should be noted that the ratio $S_p/V_p$ has the following trend, as shown in Table 1:

$$S_p/V_p \text{ CB-S} = 60.00 \, (mm^{-1}) > S_p/V_p \text{ CB-M} = 20.69 \, (mm^{-1}) > S_p/V_p \text{ CB-L} = 4.29 \, (mm^{-1})$$

As a result, the smaller the particle, the highest the surface exposed to the bulk solution. The ratio of external surface ($S_p$) per mass unit of solute ($m_{PNP}$) is:

$$S_p/m_{PNP} \text{ CB-S} = 438 \, (mm^2 \cdot mg^{-1}) > S_p/m_{PNP} \text{ CB-M} = 165 \, (mm^2 \cdot mg^{-1}) > S_p/m_{PNP} \text{ CB-L} = 34 \, (mm^2 \cdot mg^{-1})$$

These values show that the molecules of PNP have a higher extent of their area available to be adsorbed in the small particles, which explains the low values of c at $r/r_p = 1$ in sample CB-S. The PNP molecules in the smaller particles are evenly adsorbed over a high surface without saturating the carbon pores. The opposite situation takes place

in sample CB-L, giving rise to the formation of concentration gradients that hinder the mass transfer process.

### 3.2.3. Dynamic PNP Adsorption in Fixed-Bed Columns

As previously described, adsorption in a fixed-bed column is a dynamic, non-steady process, which is difficult to interpret, and the process depends on the possible interaction of several variables. To analyze the adsorption of PNP under dynamic conditions in fixed-bed columns, six experimental runs were carried out, changing the solution flow rate, the adsorbent mass and size. Table 7 shows the experimental conditions established in each run (denoted as flow-mass-size), where $\dot{V}$ is the solution flow rate, $m_c$ is the mass of carbon, $h_c$ is the column height, and $t_{contact}$ is the contact time of the liquid with the carbon bed.

**Table 7.** Parameters of the fixed-bed experiments.

|  | **(1) 11-6-S** | **(2) 5.5-6-S** | **(3) 11-3-S** | **(4) 5.5-3-S** | **(5) 11-6-M** | **(6) 5.5-6-M** |
|---|---|---|---|---|---|---|
| $\dot{V}$ (cm$^3$·min$^{-1}$) | 11.0 | 5.5 | 11.0 | 5.5 | 11.0 | 5.5 |
| $m_c$ (g) | 6.0 | 6.0 | 3.0 | 3.0 | 6.0 | 6.0 |
| $h_c$ (cm) | 5.7 | 5.7 | 2.9 | 2.9 | 5.8 | 5.8 |
| $t_{contact}$ (min) | 1.04 | 2.08 | 0.53 | 1.06 | 1.06 | 2.12 |
| $t_b$ (h) | 8.50 | 19.15 | 3.8 | 10.6 | 8.2 | 18.6 |
| $u_w$ (mm·min$^{-1}$) | $11 \times 10^{-2}$ | $5.1 \times 10^{-2}$ | $13 \times 10^{-2}$ | $5.0 \times 10^{-2}$ | $1.1 \times 10^{-1}$ | $5.1 \times 10^{-2}$ |
| $V_b/m_c$ | 0.935 | 1.017 | 0.836 | 1.056 | 0.935 | 1.034 |

This table also lists the breakthrough time ($t_b$), the wave velocity ($u_w$), and the ratio between breakthrough volume and carbon mass ($V_b/m_c$). The breakthrough curves for the six experiments have been plotted in Figure 7. The fixed-bed experiments were denoted as X-Y-Z, where X was the solution flow rate ($\dot{V} = 11$; $\dot{V} = 5.5$); Y was the mass of carbon ($m_c = 6$; $m_c = 3$); and Z was the adsorbent size (S or M).

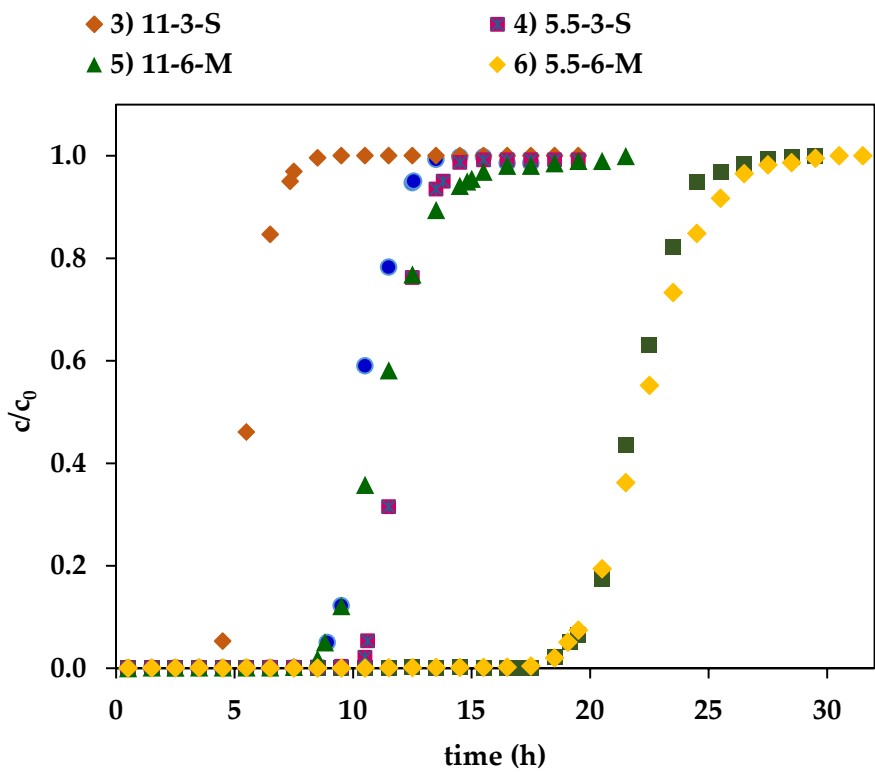

**Figure 7.** Breakthrough curves for the six experiments.

The symbols represent the experimental data obtained. It can be observed that the experimental conditions used in this study gave rise to very different patterns of the breakthrough curves. As can be observed in Figure 7, the breakthrough time decreases as follows: (2) 5.5-6-S (t = 19.15 h) > (6) 5.5-6-M (t = 18.5 h) > (4) 5.5-3-S (t = 10.6 h) > (1)11-6-S (t = 8.5 h) > (5)11-6-M (t = 8.2 h) > (3)11-3-S (t = 3.8 h).

Runs 2 and 6, which combined both lower flow rate and higher carbon mass (6g), have the highest breakthrough times. Run 3 combines both a high flow rate and low carbon mass (3 g), having a lower breakthrough time as a result. Runs 1 (11-6-S), 4 (5.5-3-S), and 5 (11-6-M) had very similar breakthrough times, although sample 4 has a slightly higher value. These results indicate that, in the experimental conditions used in this work, the inlet velocity and the mass of carbon are the processing variables that mostly determine the breakthrough time, while the particle size plays a secondary role, as mentioned above. As for the ratio $V_b/m_c$, which gives an idea of the efficiency of the column in removing contaminants, the order found is: (4) 5.5-3-S ($V_b/m_c$ = 1.056) > (6) 5.5-6-M ($V_b/m_c$ = 1.034) > (2) 5.5-6-S ($V_b/m_c$ = 1.017) > (1) 11-6-S ($V_b/m_c$ = 0.935) = (5) 11-6-M ($V_b/m_c$ = 0.935) > (3) 11-3-S ($V_b/m_c$ = 0.836).

The experiments carried out at low flow rates have the highest adsorption efficiency, with no significant differences among them despite the differences in the particle size and carbon mass used. In the runs carried out at a higher flow rate, the efficiency of the adsorption process decreased, and some differences were found observed.

It should be noted that higher flow rates imply lower contact times between the fluid and the particles. On the contrary, when the flow is slow, the contact time of the fluid with the particle is high enough for the solute to be transferred from liquid to the adsorbent in both CB-M and CB-S. Finally, run (3) 11-3-S has the lowest efficiency in the removal of solute and the lowest $t_{contact}$.

The successful design and operation of a fixed-bed column system require the production of the breakthrough curve for the effluent [16,19]. Recent studies have attempted to model the adsorption of contaminants in fixed-bed columns [40–43]. For the analysis of experimental breakthrough curves, two common models represented by Thomas [44] and Yoon–Nelson [45] were applied. The Thomas model is one of the most general and widely used theoretical methods to describe column performance [46].

This model, which assumes Langmuir kinetics of adsorption-desorption and no axial dispersion, is derived with the assumption that the rate driving force obeys second-order reversible reaction kinetics [44–48]. This model can be described by the following expression:

$$\text{Thomas model}: \quad \frac{C}{C_0} = \frac{1}{1 + \exp\left(\frac{k_{TH}}{Q}\left(q_{TH}M - C_0 V_{ef}\right)\right)} \tag{10}$$

where $k_{TH}$ (mL·h$^{-1}$mg$^{-1}$) is the Thomas rate constant, $q_{TH}$ (mg g$^{-1}$) is the adsorption capacity, $M$ (mg) is the amount of adsorbent in the column, $\dot{V}$ (mL min$^{-1}$) is the volumetric flow rate and $V_{ef}$ (mL) is the volume of effluent that has been circulating through the column at a time "t". A good correlation for breakthrough data was obtained with this model, as evident from the high values of the $R^2$ in Table 8, which range between 0.9971 and 0.9995. Thomas rate constant ($k_{TH}$) was increased with a decrease in volumetric flow rate and particle size, whereas bed depth has not a clear effect on this parameter (Table 8).

The Yoon–Nelson model is not only more simple than other models but also requires no detailed data concerning the characteristics of adsorbate, the type of adsorbent and the physical properties of the adsorption bed [49]. This model assumes that the rate of decrease in the probability of adsorption for each adsorbate molecule is proportional to the probability of adsorbate adsorption and the probability of adsorbate breakthrough on the adsorbent [45,48]. The Yoon–Nelson equation for a single-component system is expressed as

$$\text{Yoon–Nelson model}: \frac{C}{C_0} = \frac{1}{1 + \exp(k_{YN} \cdot (\tau - t))} \tag{11}$$

where $k_{YN}$ (h$^{-1}$) is Yoon–Nelson rate constant, $\tau$ (h) is the time required for 50% adsorbate breakthrough, and $t$ is time (h). A good correlation between the experimental data to the Yoon–Nelson model was observed (Table 8). The time required for 50% breakthrough values ($\tau$) was decreased with increasing flow rate and, obviously, increased with the rise in bed height (Table 8).

**Table 8.** Thomas, Yoon–Nelson, and 2D models parameters for PNP adsorption for the six experiments carried out.

| | Run 1 11-6-S | Run 2 5.5-6-S | Run 3 11-3-S | Run 4 5.5-3-S | Run 5 11-6-M | Run 6 5.5-6-M |
|---|---|---|---|---|---|---|
| $k_{TH}$ mL (h$^{-1}$mg$^{-1}$) | $0.145 \pm 0.010$ | $0.171 \pm 0.003$ | $0.172 \pm 0.005$ | $0.172 \pm 0.005$ | $0.0905 \pm 0.002$ | $0.143 \pm 0.005$ |
| $q_{TH}$ exp. (mg·g$^{-1}$) | 222.8 | 234.5 | 254.1 | 254.1 | 240.7 | 233.4 |
| $q_{TH}$ cal. (mg·g$^{-1}$) | $230.2 \pm 1.1$ | $241.0 \pm 0.3$ | $261.7 \pm 0.3$ | $261.7 \pm 0.3$ | $244.8 \pm 1.0$ | $281.4 \pm 1.1$ |
| $R^2$ | 0.9971 | 0.9993 | 0.9995 | 0.9995 | 0.9990 | 0.9980 |
| $k_{YN}$ (h$^{-1}$) | $1.67 \pm 0.12$ | $1.67 \pm 0.12$ | $2.07 \pm 0.18$ | $2.07 \pm 0.18$ | $1.09 \pm 0.28$ | $1.66 \pm 0.19$ |
| $\tau$ exp. (h) | 10.28 | 21.82 | 5.60 | 11.92 | 11.13 | 22.23 |
| $\tau$ cal. (h) | $10.46 \pm 0.05$ | $21.91 \pm 0.24$ | $5.61 \pm 0.30$ | $11.90 \pm 0.34$ | $11.13 \pm 0.65$ | $27.95 \pm 0.65$ |
| $R^2$ | 0.9970 | 0.9992 | 0.9996 | 0.9996 | 0.9990 | 0.9980 |
| $t_b$ (h) | 8.84 | 18.7 | 3.50 | 10.33 | 8.00 | 18.08 |
| $z_{w,aveg}$ | 1.51 | 1.23 | 1.77 | 0.95 | 1.43 | 1.10 |
| $R^2$ | 0.9824 | 0.989 | 0.9897 | 0.9864 | 0.9949 | 0.9986 |

In summary, Thomas and Yoon–Nelson models exhibited a good correlation with the experimental breakthrough data for the six samples. Hence, these models can be used to describe the adsorption performance of PNP in a fixed-bed column. Nevertheless, these models are empirical, and their applicability is limited because they do not consider both the geometry of the system and the transport phenomena (mass, momentum, and heat) occurring during the treatment. For this reason, we also applied a 2D model to describe the adsorption of a solute in a fixed-bed column of adsorbent, taking into consideration the velocity of the fluid through the channel bed. In Figure 8, the simplified model used has been represented. As it can be observed, a unique ideal channel has been analyzed, in which the flowing fluid is surrounded by half spheres of adsorbent.

The length of the channel is that of the column, and the fraction of the system area occupied by the fluid was set to be equal to the bulk porosity of the column. The Navier–Stokes equation was the equation used to describe the fluid motion [50]:

$$\rho \frac{\partial u}{\partial t} + \rho(u \cdot \nabla)u = \nabla \cdot \left[ -pI + \mu\left(\nabla u + (\nabla u)^T\right) - \frac{2}{3}\mu(\nabla \cdot u)I \right] + F \tag{12}$$

where $\rho$ is the bulk density (kg·m$^{-3}$), $u$ is the velocity (m·s$^{-1}$), p is the pressure (Pa), I is the unit matrix, $\mu$ is the viscosity (Pa·s), $\nabla$ is the gradient (m$^{-1}$), $(\nabla u)^T$ is the transpose of the velocity gradient, and F is the volume force (N·m$^{-3}$).

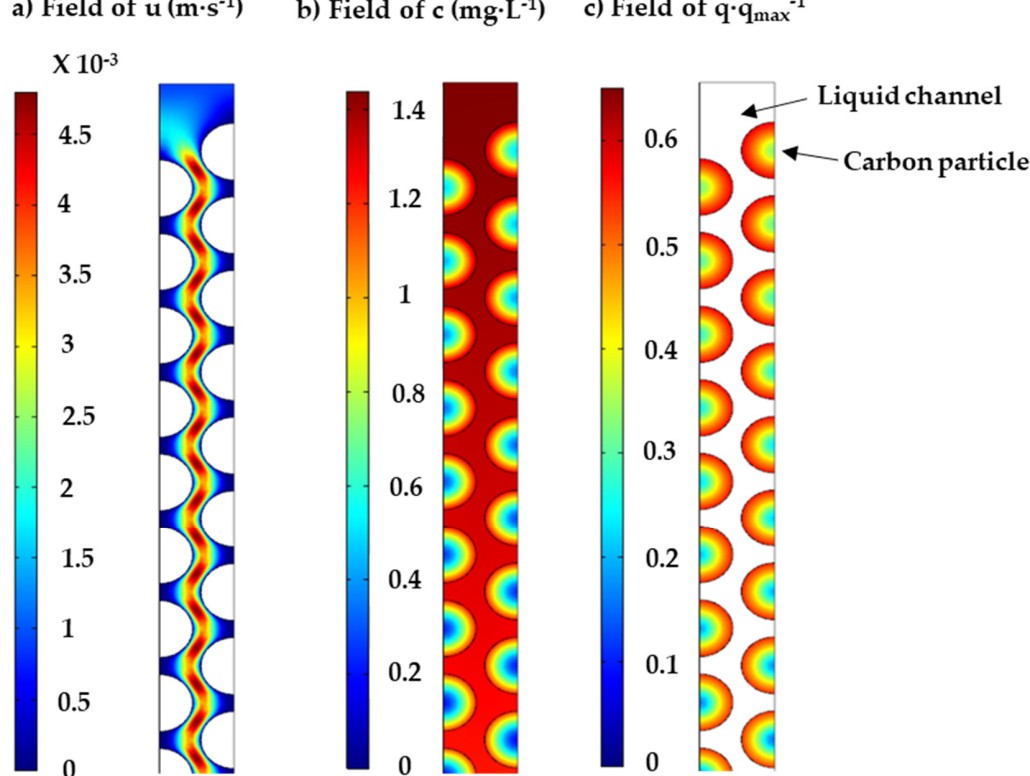

**Figure 8.** Scheme of the 2D model for the column and some of the fields calculated determined by the model: (**a**) velocity (u), (**b**) concentration of solute dissolved (c), and (**c**) ratio (q·q$_{max}$$^{-1}$).

Assuming the liquid is incompressible, this involves that the density is constant ($\rho$) and that the divergence of the velocity is zero:

$$\rho \nabla \cdot u = 0 \tag{13}$$

Moreover, if it assumed that the liquid flows under steady-flow conditions and is not affected by gravity, the Navier–Stokes equation is simplified as:

$$\rho(u \cdot \nabla)u = \nabla \cdot \left[ -\mathrm{pI} + \mu \left( \nabla u + (\nabla u)^{\mathrm{T}} \right) \right] \tag{14}$$

The density of the solution was set to 103 kg·m$^{-3}$ because the concentration of solute was low. With these assumptions, Equation (14) was resolved with COMSOL Multiphysics, taking into account that the liquid was flowing at steady-read. Under these flowing conditions, the field u can be considered as a "fixed picture" (that is, independent of time), which provides that the interstitial velocity u at any point of the system is valid for the whole experiment, u (x, y, z). This enormously simplifies the computational work required. The plot in Figure 8 shows an example of this field; it can be observed that the maximum interstitial velocity takes place in the center of the channel. Once the field of u has been found, the concentration of solute dissolved (c) and adsorbed (q) can be calculated. The equations used for defining the mass transfer of solute in the interphase and through the particle were the same as those employed for the batch model (Equations (7)–(9)), while the field of c in the flowing fluid was:

$$\frac{\partial c}{\partial t} + \nabla(-D_{F,l} \nabla c) + u \cdot \nabla c = 0 \tag{15}$$

where $D_{F,l}$ is the PNP diffusion coefficient in the liquid fluid flowing through the channel, which was set equal to $5 \times 10^{-9}$ (m·s$^{-1}$) for all the experiments. Equation (15)

states that the mass of solute accumulated in a volume element (first term) is the balance between the mass going in and out by diffusion (second term) and convection (third term). Equations (7)–(9) and (15) are dependent on time and were solved simultaneously with COMSOL Multiphysics, using the $u$ field provided by the Navier–Stokes equation (Equation (14)). The model gives the solute dissolved and adsorbed at any point of the system and at any time of the experiment, c (x, y, z, t) and q (x, y, z, t), respectively. Figure 8 shows an example of the field of $c$ and $q \cdot q_{max}^{-1}$ at a given time.

The inputs in the model for describing the adsorption in the fixed bed can be grouped as follows:

(1) Particle parameters: the size, porosity, and weight of the particle (Table 1).
(2) Mass transfer and isotherm parameters: the two Langmuir parameters (b and $q_{max}$), the constant of proportionality in the interphase, and the diffusion coefficient into the particle. The values of these four parameters were those provided by the batch study (Table 6).
(3) Flow and bed parameters: flow rate, carbon mass, diameter, and length of the bed (Table 7).

In Figure S5, the model values of the breakthrough curve are represented as red lines. From this figure and the high values of $R^2$ (Table 8), it is evident that the model accurately fits the breakthrough curve in all experiments, giving values of breakthrough time ($t_b$) like the experimental ones. The real value of the model is that it allows performing a deeper analysis of the adsorption process. According to the wave theory [15], during the dynamic adsorption in a column, there is a wave of concentration advancing through the column. The adsorbent bed can be divided into three parts with different solute concentrations in the solid and liquid phases (q and c, respectively):

(1) Beyond the wave, loaded with the pristine carbon (q = 0, c = 0),
(2) Wave, where the mass transfer is taking place, and both q and c are changing,
(3) Post-wave zone, in which c is the concentration entering the column ($c_0$) and the carbon has a constant contaminant load in equilibrium with $c_0(q_e)$.

By using the 2D model, it is possible to study how the shape of the concentration wave evolves during its flow through the column. Figures 9 and S6–S10 show the wave at different times for each experiment, as well as the wavelength ($z_w$), considering that the end of the wave corresponded with a $c/c_0$ value of 0.95.

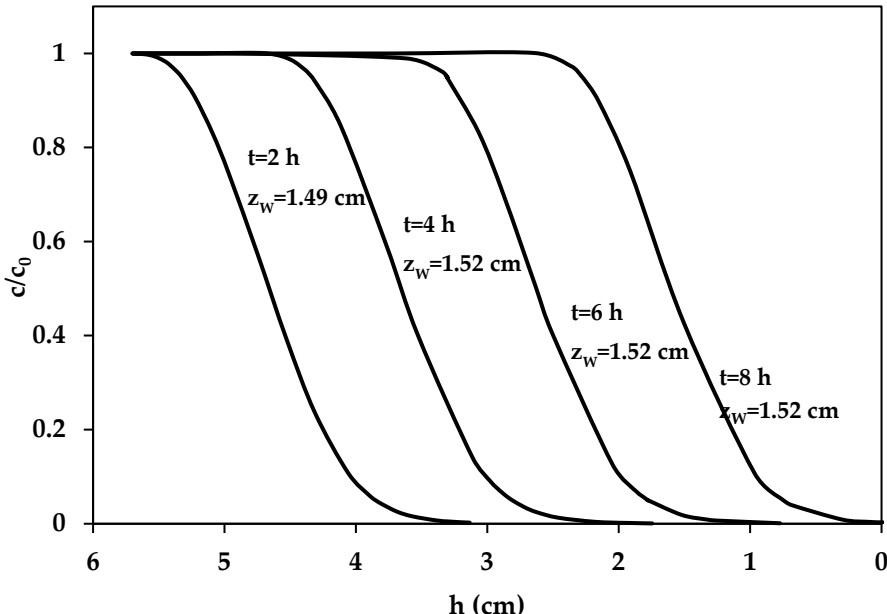

**Figure 9.** PNP concentration wave at 2 h, 4 h, 6 h, and 8 h for the run (1) 11-6-S.

It can be observed that the shape and the wavelength of the wave for each run remain almost constant most of the way. This pattern of behavior can be explained as follows. According to the equilibrium theory, the wave velocity ($u_w$) can be expressed as:

$$u_w = \frac{u}{1 + \left(\dfrac{\rho}{\varepsilon}\right)\left(\dfrac{d_q}{d_c}\right)} \tag{16}$$

As $d_q/d_c$ is the slope of the equilibrium isotherm at the corresponding c concentration, this equation shows that the wave velocity depends on the isotherm type. The PNP adsorption on CB-L, CB-M, and CB-S have a Langmuir-type isotherm (convex, favorable isotherm), which is a slope described in Equation (1). The advancing front of the wave has a high slope and, consequently, $u_w$ will be low. On the contrary, the rear part of the wave has a concentration near to $c_0$ and the wave velocity would be like the fluid velocity ($u_w \approx u$). As a result, the rear part of the wave will be faster than the front, which explains the wave sharpening at the initial time. As the waves travel through the column, the dispersive effect of the non-idealities prevents the wave from sharpening into a discontinuity (the rear part overtaking the front). The main non-idealities in fixed-bed adsorption tending to spread the wave are (1) deviation from local equilibrium due to finite rate of equilibration between the water stream and the carbon bed; (2) axial diffusion; and (3) non-plug flow. The wave profile will change until a finite width is reached, for which the balance between sharpening and spreading tendencies is attained. Then the wave will travel without further change and is said to attain a Constant Pattern. The average wavelength ($z_{w,aveg}$) for each experiment is listed in Table 8. Under the experimental conditions used in this study, this parameter ranged between 0.95 and 1.77 and increased with a rising flow rate. It is interesting to note that there is a negative linear relationship between $z_{w,aveg}$ and $V_{tb}/m_c$, that is, between the wavelength and the efficiency of the adsorption process in the column bed:

$$\frac{V_{tb}}{m_c} = -0.271 \cdot z_w + 1.331 \qquad R^2 = 0.9655 \tag{17}$$

From a practical point of view, it is interesting to note that all the parameters provided by the equilibrium and batch studies allow a good fitting of the experimental data of the fix-bed column adsorption only by considering the field of velocity in the system. In other words, with the information provided by the characterization of the solid (geometry, He pycnometry, and Hg porosimetry) and the batch model, the 2D model adequately predicts the adsorption of the complex adsorption process that takes place in a fix-bed carbon column. This result is very interesting because it indicates that the model could be used to predict the adsorption process in carbon filters in water-treatment plants by describing the flow in the fixed bed of the filter.

## 4. Conclusions

The three samples used in this study (CB-L, CB-M, and CB-S) showed a very similar porosity pattern. Their values of microporosity ($V_{mi}$), mesoporosity ($V_{me}$), particle density ($\rho_p$) and true density ($\rho_t$) were very similar. On the contrary, they had different sizes and, consequently, different particle surfaces ($S_p$), particle volume ($V_p$), and the ratio $S_p/V_p$.

The three adsorption isotherms were very similar and can be classified as type L according to the Giles classification. The adsorption isotherms have been adjusted by the Langmuir and Double-Langmuir models and Dubinin–Radushkevich equation. And all models demonstrate excellent fit to the data.

The three samples showed different behavior in the adsorption experiment carried out in batch conditions, despite their similarities in porosity. The smaller the particle, the faster the adsorption process. A 2D model that properly described the evolution of both c and q was developed. The model allows one to visualize the changes of c and q inside the particle, helping the analysis of the experimental results. The sample with the largest particles (CB-L) had important gradients of q. The model predicts that this behavior is due

to the high values of c in the outer part of the particle. This high value of c brings about a high value of q and a hindrance to the mass transfer from the bulk liquid to the solid. As the diffusion of the particle is a slow process, the inner part of the particle is free of solute. The result of these facts is the formation of a gradient of q in the particle. The other samples have smaller particles, and due to the high $S_p/V_p$ ratio, the values of c at the outer part of the particle are small. As a result, there is no important gradient in samples CB-M and CB-S.

A simplified 2D model was developed to describe the adsorption process in a fixed-bed column. The model calculates the velocity in the system by applying the Navier–Stokes equation. The only input used were those provided by the characterization of the solids, the isotherms, and the batch adsorption model. The model can be useful for studying the adsorption in filters used for the removal of contaminants.

The results presented in this paper show that with the combined use of simple characterization analyses and the two 2D models proposed, it is possible to describe the adsorption process in any system, providing a useful tool in adsorption applications on both lab and industrial scales.

**Supplementary Materials:** The following supporting information can be downloaded at: https://www.mdpi.com/article/10.3390/pr11072045/s1, Figure S1: Experimental and 2D-Batch model kinetic values of PNP concentration in the solutions, c; Figure S2: Experimental and 2D-Batch model kinetic values of PNP concentration in the carbons, q; Figure S3: Experimental and PFO model kinetic values of PNP concentration in the carbons, q; Figure S4: Experimental and PSO model kinetic values of PNP concentration in the carbons, q; Figure S5: 2D-column breakthrough curves for the six experiments; Figure S6: PNP concentration wave at 4 h, 8 h, 12 h, and 16 h for the run (2) 5.5-6-S; Figure S7: PNP concentration wave at 1 h, 2 h, and 3 h, for the run (3) 11-3-S; Figure S8: PNP concentration wave at 2 h, 4 h, 6 h, and 8 h, for the run (4) 5.5-3-S; Figure S9: PNP concentration wave at 2 h, 4 h, 6 h, and 8 h, for the run (5) 11-6-M; Figure S10: PNP concentration wave at 4 h, 8 h, 12 h and 16 h, for the run (6) 5.5-6-M.

**Author Contributions:** B.L.: conceptualization, methodology, investigation, writing–original draft, and writing—review and editing; E.S.: conceptualization, software, validation, formal analysis, and writing–review and editing; C.M.G.-G.: conceptualization, software, and writing—review and editing; S.R.: methodology, investigation, and writing—original draft; M.E.F.: methodology, writing—review and editing, and supervision; P.B.: methodology, writing—review and editing, and supervision; A.L.C.: methodology, writing—review and editing, and supervision. All authors have read and agreed to the published version of the manuscript.

**Funding:** This research was funded by Agencia Estatal de Investigación (MINCIN), grant number PID2020-116144RB-I00/AEI/10.13039/501100011033XXX.

**Data Availability Statement:** Not applicable.

**Acknowledgments:** The authors thank the SAIUEX (Servicios de Apoyo a la Investigación de la Universidad de Extremadura) for support during the characterization of materials.

**Conflicts of Interest:** The authors declare no conflict of interest. The funders had no role in the design of the study; in the collection, analyses, or interpretation of data; in the writing of the manuscript; or in the decision to publish the results.

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
