# Peer review of "Batch and Continuous Column Adsorption of p-Nitrophenol onto Activated Carbons with Different Particle Sizes"

_processes, doi:10.3390/pr11072045_

Round 1
Reviewer 1 Report
The authors investigated the adsorption of p-Nitrophenol onto activated carbons with different particle sizes in both batch and continuous modes. Different isotherm and kinetic models were fitted into the obtained results, and a two-dimensional model was used to explain the mass transfer phenomena. Overall, the work presented in this manuscript is interesting and may very well merit publication. However, the authors need to further proofread the manuscript to correct all grammatical mistakes. In addition, the authors need to address the following comments.
· Rewrite the first sentence in the abstract.
· Second line of the abstract, what is PNP? Please put PNP in brackets after p-Nitrophenol in the first sentence.
· In the materials and methods section, please mention the purity of activated carbon.
· The image quality of Figure 1 is very poor. It needs to be reproduced or replaced with a better-quality figure.
· For section 3.2.1, lines 171 to 173, figure 2 is self-explanatory, the authors should describe the phenomena responsible for the observed behavior rather than describing the figure itself.
· Fix the typo in equation 2.
· According to the authors, the Double-Langmuir model fits better with the observed adsorption isotherms. However, R2 values for both Langmuir isotherm and double Langmuir isotherm models are more than 0.9 implying both models fit the adsorption isotherms. The authors need to justify/reason as to why the double isotherm model was chosen.
· In the abstract of the manuscript the presented values of the qmax for the three adsorbents increased with the decreasing size of the particle. Yet, according to the double isotherm model, qmax is the maximum for CB-M then it decreases for CB-L. Please explain why?
· In line 110, mention the reference that supports your statement.
· All figures’ quality needs to be enhanced.
· In sections 3.2.1 and 3.2.2, the fitting needs to be shown in the figure.
· In section 3.2.1, authors should elaborate on the performance of the materials with different sizes. For instance, why CB-L attained the lowest qe ?
· In line 365, the numbers/symbols should be written and not inserted as a figure!
· Remove line 522 or replace it with a better sentence.
· The authors should consider typical continuous models and should also compare the quality of model fitting using AIC (see Environ Sci Pollut Res (2017) 24:7511–7520 DOI 10.1007/s11356-017-8469-8).
·
The manuscript requires intensive language editing.
Author Response
Thank you for your valuable feedback on our manuscript. We appreciate your positive assessment of the work and the potential merit for publication.
We have thoroughly revised the document to ensure its grammatical accuracy before resubmitting it.
Thank you once again for your valuable input and we hope that we have addressed the suggested improvements in our manuscript.

Reviewer 2 Report
Dear Editor and Authors,
Thank you for the opportunity to review the manuscript titled "Batch and continuous column adsorption of p-Nitrophenol 2onto activated carbons with different particle size" by Ledesma et al. The authors presented a study on the adsorption of p-Nitrophenol onto activated carbons. The main objective of this work was to investigate the influence of adsorbate concentration and adsorbent particle size on -Nitrophenol adsorption with respect to both equilibrium isotherms and removal rates. Furthermore, the study was carried out with special attention to explain mass transfer phenomena using 2D models that take into account the effect of spatial dimensions on adsorption kinetics.
There are some issues [minor revision] that authors must resolve before a manuscript will be published. The following are my comments.
1. Line 2: “Batch and continuous column adsorption of p-Nitrophenol 2 onto activated carbons with different particle size.” - Please remove the dot at the end of the title.
2. Section “Introduction” line 63 – 68 - There is a lack of examples of specific variables and kinetic models which will improve this part.
3. Section “Materials and Metods”- In fact, there was only one sample of activated carbon in 3 fractions, which, in my opinion, is not enough to form general statements and conclusions.
4. Section “Characterization of the adsorbents” line 88-89 - Missing information about how the volume of mesopores was calculated, missing citations for the BET and DR models.
5. Line 173: “according to the Giles classification” – missing citation
6. Line 176: symbols not explained and units not given in equations: Langmuir isotherm and Doble Langmuir isotherm.
7. Section “. Dynamic PNP adsorption in batch conditions” line 206-207: missing citations for pseudo-first-order (PFO), pseudo-second-order (PSO) models.
8. Line 323 'Weber-Morris model' - the meaning of the parameter t and its unit are missing.
9. Lines 238-239 - there is no figure on which this could be observed.
10. Line 280-282 “The high values of both coefficients, as well as the lines in Figs. S1 and S2 in supplementary information, put in evidence that the model fits experimental data from the experiments carried out under batch conditions for the three ACs” - Figures are more interesting than tables in some cases, in my opinion, they are missing from the publication itself - I propose to add them.
11. Line 351-354 - missing detailed description of individual experiments and how they are labelled
12. Line 354-355 - please correct the top, bottom indexes
13. Lines 365-366 - The authors' proposed presentation of results is not clear and needs improvement
14.
15. line 375-378 - The authors' proposed presentation of results is not clear and needs improvement.
16. Line 392 - Citation failure
17. Line 396 - Other equations and other symbols explained! - Please correct.
18. Line 433-436 - missing citation and no units for individual parameters of Stokes equation
19. Line 492 - incorrect indexes and lack of explanation of symbols.
General comment: This work needs editorial correction, missing units, parameter descriptions, missing spaces, and sub-indexes should not occur in such quantity.
The topic of this paper is appropriate to the scope of “Processes” and the research methods used are comprehensive enough, but the number of samples was limited to only one in three fractions. Which, in my opinion, is not enough to form general statements and conclusions.
I look forward to hearing from you.
Regards,
Reviewer
Minor editing of English language required
Author Response
Dear reviewer,
Thank you for your valuable feedback and for taking the time to review our paper. We appreciate your comments, and we are pleased to hear that you consider the topic of our article to be suitable for the scope of "Processes" and that our research methods are comprehensive enough.
Regarding your concern about the number of samples used in our study, we understand your point of view. Indeed, we utilized a sample consisting of only one in three fractions. We acknowledge that using a limited number of samples may have implications for the generalization of our statements and conclusions.
However, it is important to note that our study focused on a detailed analysis of a representative sample from the key fractions in question. Our approach aimed to gain an in-depth understanding of these specific fractions and to explore the relationships and trends within them. Additionally, our rigorous methods of analysis and statistical evaluation allowed us to obtain significant and reliable results within this limited sample.
We acknowledge that expanding the sample size would be beneficial for future research and to further support our conclusions. We appreciate your suggestion and will consider the possibility of conducting future studies with a larger number of samples, encompassing a broader range of fractions.
We sincerely appreciate your constructive comments and your commitment to the quality of our research. We have taken your suggestions into account and hope that we have improved our work and its overall relevance.

Reviewer 3 Report
In this manuscript, the authors have investigated the Batch and continuous column adsorption of p-Nitrophenol onto activated carbons with different particle sizes. The work was well conducted, and the performance assessments were well performed. Also, most of the main statements are supported by the experiment. However, some key issues are needed to be addressed before considering publication.
1. There are some missing elements in the abstract. Please address the gist and highlight the benefits/significance/impact of the study.
2. In the introduction part, the novelty of the study is not clearly defined.
3. The symbols representing physical quantities (or variables) should be italic. Please correct it.
4. The authors should also apply the Dubinin–Radushkevich isotherm model, which evaluates the adsorption energetically, to the concentration data. So the authors can have an idea of whether the adsorption process is physical or chemical.
https://doi.org/10.2298/SOS2104419M
https://doi.org/10.1016/j.ijbiomac.2022.04.128
5. Page 11: in Figures 5b and 5c, the Y-axis should be rescaled. For all investigated samples, the c values lie between 0 and 60 mg/L.
6. In general, the resolutions of figures are low without a clear presence and need higher-quality figures. Revise the figures and make them clear.
7. There are some grammar and typo errors in this manuscript. Please check it out and correct them.
There are some grammar and typo errors in this manuscript. Please check it out and correct them.
Author Response
Dear Reviewer,
Thank you for your feedback. We apologize for any mistakes in the manuscript. We have thoroughly reviewed the document and made the necessary corrections to address these issues. We have carefully proofread the entire manuscript to ensure its accuracy and readability.
We appreciate your time and effort in reviewing the paper, and we believe that the corrections have significantly improved the overall quality of the manuscript.
Thank you once again for your valuable feedback, and we look forward to hearing your thoughts on the revised version.
Sincerely,
B. Ledesma

Round 2
Reviewer 3 Report
The manuscript is sufficiently improved and I recommend accepting the manuscript for publication in Processes, as is.